# Comparative Analysis of Intestinal Inflammation and Microbiota Dysbiosis of LPS-Challenged Piglets between Different Breeds

**DOI:** 10.3390/ani14050665

**Published:** 2024-02-20

**Authors:** Chao Li, Yanping Wang, Xueyan Zhao, Jingxuan Li, Huaizhong Wang, Yifan Ren, Houwei Sun, Xiaodong Zhu, Qinye Song, Jiying Wang

**Affiliations:** 1Shandong Key Laboratory of Animal Disease Control and Breeding, Institute of Animal Science and Veterinary Medicine, Shandong Academy of Agricultural Sciences, Jinan 250100, China; 18253829358@163.com (C.L.);; 2Key Laboratory of Livestock and Poultry Multi-Omics of MARA, Jinan 250100, China; 3Hebei Veterinary Biotechnology Innovation Center, College of Veterinary Medicine, Hebei Agricultural University, Baoding 071000, China; 4Zaozhuang Heigai Pigs Breeding Co., Ltd., Zaozhuang 277100, China

**Keywords:** Heigai pigs, intestinal inflammation, jejunum microbiota, lipopolysaccharide, post-weaning diarrhea, 16S rDNA-Seq

## Abstract

**Simple Summary:**

Post-weaning diarrhea is common in piglets, causing huge economic losses worldwide. The Heigai pig is an indigenous pig breed of China, and has lower rate of piglet diarrhea incidences than the Duroc × Landrace × Yorkshire (DLY) crossbred pig, which is the most widely used commercial pig crossbreed worldwide. In the current study, using lipopolysaccharide (LPS) as an agent to mimic the bacterial inflammatory response in weaned pigs, a comparative analysis of the effects of LPS challenge on jejunal mucosal morphology, jejunal microbial composition, and serum indexes in two pig breeds (DLY and Heigai) was performed. The results showed that LPS successfully induced bacterial inflammatory response in the experimental pigs, changed the microbial composition, and triggered functional changes in energy metabolism and activities related to the stress response in the jejunal bacterial community. This study also revealed that Heigai pigs had a weaker immune response to LPS challenge than DLY pigs, and identified several genera related to the breed-specific phenotypes of Heigai pigs. These findings provide a basis for investigating the mechanism of the intestinal inflammatory response in piglet diarrhea and suggest potential microbial candidates to resist diarrhea.

**Abstract:**

Post-weaning diarrhea is common in piglets, causing huge economic losses worldwide. Associations between LPS challenge, intestinal inflammation, and microbiota have been reported in Duroc × Landrace × Yorkshire (DLY) crossbred pigs. However, the effects of LPS challenge in other breeds remain unclear. In the current study, we performed a comprehensive comparative analysis of the effects of LPS challenge on jejunal mucosal morphology, jejunal microbial composition, and serum indexes in two pig breeds: DLY and Heigai, an indigenous Chinese breed. The results showed that LPS caused considerable damage to the mucosal morphology, enhanced serum levels of inflammatory cytokines and the intestinal permeability index, and lowered the antioxidant capacity index. LPS challenge also changed the microbial composition and structure of the jejunum, significantly increased the abundances of *Escherichia-Shigella* in DLY pigs, and decreased those of *Gemella* and *Saccharimonadales* in Heigai pigs. Furthermore, LPS challenge triggered functional changes in energy metabolism and activities related to the stress response in the jejunal bacterial community, alleviating the inflammatory response in Heigai pigs. This study also revealed that Heigai pigs had a weaker immune response to LPS challenge than DLY pigs, and identified several genera related to the breed-specific phenotypes of Heigai pigs, including *Gemella*, *Saccharimonadales*, *Clostridia_UCG_014*, *Terrisporobacter*, and *Dielma*. Our collective findings uncovered differences between Heigai and DLY pigs in intestinal inflammation and microbiota dysbiosis induced by LPS challenge, providing a theoretical basis for unraveling the mechanism of intestinal inflammation in swine and proposing microbial candidates involved in the resistance to diarrhea in piglets.

## 1. Introduction

The mammalian intestinal tract plays a crucial role in many important processes, such as digestion, nutrient absorption, and immune function [1]. It houses a diverse community of microorganisms, many of which perform essential functions for host physiology, including providing microbial by-products as nutrients and contributing to immune regulation [2,3]. However, pathogenic infection, stressors (e.g., cold, weaning, abrupt changes in the diet, and environment) and other factors in pig production often induce intestinal inflammation and cause intestinal mucosal injury and microbiota dysbiosis, resulting in poor growth of pigs and even sudden death of piglets. The weaning phase is an especially critical window for gut microbiota formation [4,5]. Weaning piglets, whose intestinal microbiota is quite vulnerable to weaning stress, are highly susceptible to external stimuli. Understanding the biological impact of weaning stress is crucial for improving strategies to overcome it.

Diarrhea in the first two weeks after weaning is a common illness in pig production worldwide and causes huge economic losses. It is usually associated with the proliferation of enterotoxigenic *Escherichia coli* in the gastrointestinal tract [6]. Lipopolysaccharide (LPS), also termed endotoxin, is a major component of the outer membrane of Gram-negative bacteria, and has three regions: the conserved lipid A anchor, core oligosaccharide, and O antigen [7]. It is also one of the most effective stimulators of the immune system, inducing strong inflammatory responses and various biological effects [8,9]. LPS has a wide range of specific applications for the study of inflammation in both in vitro and in vivo models because it stimulated many of the inflammatory effects of cytokines, including tumor necrosis factor (TNF)-α, interferon (INF)-γ, and interleukin (IL)-10 [10,11].

China has a large number of indigenous pig breeds formed by long-term natural and artificial selection, many of which exhibit distinct characteristics, such as desirable meat quality [12], high resistance to disease [13,14,15], and good adaptability to roughage [16]. Recently, extensive studies on the intestinal microbiota of indigenous pig breeds revealed that the composition and proportion of major bacterial phyla and genera of intestinal microbiota varied among pig breeds and contributed to their phenotypic formation [13,14,17,18], providing a new avenue for investigating breed-specific variation in susceptibility/resistance to post-weaning diarrhea. Heigai is an indigenous fatty pig breed and has specific characteristics like strong disease resistance and strong adaptability to the environment, and is mainly distributed in the Shandong province of China [19]. In production, a lower diarrhea rate of weaned piglets was observed in Heigai than in Duroc × Landrace × Yorkshire (DLY) crossbred (personal communication, March 2022). However, the mechanism of resistance to post-weaning diarrhea in Heigai pigs has not been reported. We hypothesized that the distinct gut microbiota in Heigai pigs may partially explain this phenotypic feature.

In the current study, we aimed to characterize and compare the gut microbiota and screen for unique bacteria involved in the diarrhea response in Heigai and DLY piglets, using an intestinal inflammation model established through intraperitoneal injection of LPS. Changes in gut morphology, serum index, and microbiota were measured to analyze differences in the inflammatory reaction and intestinal microbial alterations in response to LPS challenge between the two breeds. This study aims to apply an LPS-induced inflammation model to investigate and compare the breed-specific factors leading to intestinal inflammation in piglets, and may contribute to future screening for intestinal microbes associated with this common illness.

## 2. Material and Method

### 2.1. Animals and Experimental Treatments

A total of 20 healthy, 28-day-old weaned piglets from Heigai and DLY pigs (n = 10 per breed) were raised in a common environment at Zaozhuang Heigai Pigs Breeding Co., Ltd. (Shandong, China), and the standard starter feed of early weaning and clean water were available ad libitum, with 10 pigs as a group fed separately in a stainless-steel cage (2.00 m × 2.00 m × 1.00 m). The nutrient and energy concentrations of the experimental diets are presented in Table 1, and the nutrient levels of the diets were formulated to meet or exceed the nutritional needs of pigs according to the NRC (2012). The piglets were not administered any probiotics or antibiotics throughout the entire duration of the study. The temperature in the pigpen was regulated to stay within the range of 20 to 25 °C during the experiment. Seven days after weaning, the piglets were randomly divided into four groups: LPS-treated DLY pigs (DLY.LPS; sample ID D1-D5); untreated control DLY pigs (DLY.CON; sample ID D6-D10); LPS-treated Heigai pigs (HG.LPS; sample ID H1-H5); and untreated control Heigai pigs (HG.CON; sample ID H6-H10). Detailed information on the experimental piglets in each group is presented in Appendix A. The two LPS groups received an intraperitoneal injection of LPS (*E. coli* serotype 055: B5, Sigma Chemical, St. Louis, MO, USA) at a dose of 100 μg/kg body weight as previously described [20], while the two control groups received an intraperitoneal injection of an equal volume of 0.9% (*w*/*v*) NaCl solution. After injection of LPS or NaCl, the experimental pigs were kept in quiet environment and carefully observed to record their response.

### 2.2. Sample Collection and Processing

Blood samples were obtained from piglets in the supine position by puncture of the anterior vena cava after 4 h of treatment with LPS or NaCl solution. The collected blood samples were centrifuged at 2200× *g* for 5 min to obtain serum, which was stored at −20 °C for further analysis. Piglets were sacrificed immediately after blood sampling, and sections of the mid-jejunum were sampled from each individual and fixed in 4% paraformaldehyde. Fresh jejunal contents were collected from each individual in sterile plastic tubes and immediately snap frozen in liquid nitrogen, and then stored at −80 °C until DNA extraction.

### 2.3. Intestinal Morphology

For histological analysis of the mid-jejunum, formalin-fixed and paraffin-embedded tissues were sectioned into thick (5 μm) slices, which were then stained with hematoxylin and eosin (H&E) for further examination.

Complete histological sections were scanned using a digital pathology section scanner (Hamamatsu Photonics, Hamamatsu, Japan) to enable digital imaging. Villus height and crypt depth were measured from digital images using NDP View software (NDP View 2).

### 2.4. Determination of Serum Indexes

Serum diamine oxidase (DAO) activity was measured using an assay kit (Jiancheng Institute of Bioengineering, Nanjing, China). Porcine enzyme-linked immunosorbent assay (ELISA) kits were used to measure IL-10, TNF-α, IFN-γ (CUSABIO Biotechnology Co., Ltd., Wuhan, China), and immunoglobulin (Ig) A (Solarbio Biotechnology Co., Ltd., Beijing, China). IgG, IgM, and superoxide dismutase (SOD) were determined by an autoanalyzer of Hitachi-7100 (Hitachi, Tokyo, Japan) using the corresponding kits (MedicalSystem Biotechnology Co., Ltd., Ningbo, China).

### 2.5. DNA Extraction and Sequencing Analysis

A MagPure Stool DNA LQ Kit (Majorbio Bio-Pharm Technology Co., Ltd., Shanghai, China) was used to extract genomic DNA from jejunal contents following the manufacturer’s instructions. The concentration and purity of the extracted DNA were measured using a NanoDrop ND-2000 Spectrophotometer (Thermo Scientific, Waltham, MA, USA), and the integrity of the extracted DNA was assessed using a 1% (*w*/*v*) agarose gel. The V3-V4 region of the 16S rRNA gene was amplified using primers 341F (CCTACGGGNGGCWGCAG) and 805R (GACTACHVGGGTATCTAATCC), as reported previously [21]. PCR reactions were performed under the following cycling conditions: initial denaturation at 95 °C for 3 min; 25 cycles of denaturation at 95 °C for 30 s, annealing at 55 °C for 30 s, and extension at 72 °C for 15 s; and a final extension at 72 °C for 5 min. PCR products were purified using magnetic beads, and quantitative analyses were performed using a Qubit^®^ RNA Assay Kit (Thermo Fisher Scientific, Waltham, CA, USA). The sizes of the PCR products were checked on a 1.5% agarose gel. To generate the libraries, quantified PCR products were purified and adjusted to equal amounts for each sample using a TruSeq Nano DNA LT Library Prep Kit (Illumina Inc., San Diego, CA, USA). These libraries were then subjected to paired-end sequencing on a Novaseq PE250 platform (Illumina Inc.) by Benagen Ltd. (Wuhan, China).

### 2.6. Amplicon Sequence Variant (ASV) Annotation

The raw sequencing data were initially processed using Trimmomatic v0.39 [22] to remove low-quality reads and Cutadapt v3.5 [23] to recognize and remove primer sequences. Next, DADA2 [24] was employed to filter and trim, learn the error rates, merge paired reads, infer the ASV table based on the core sample inference algorithm, and remove sequence chimeras. Finally, taxonomic assignment of ASVs was performed using QIIME2 (version 2022.3) [25], based on the SILVA [26] 138 release (https://www.arb-silva.de/documentation/release-138/) (accessed on 1 September 2022). Abundance differences of bacteria at taxonomic levels of phylum and genus were assessed by Wilcoxon rank sum test, with *p* value < 0.05 considered to indicate statistical significance.

### 2.7. Alpha and Beta Diversity Analysis

Alpha diversity analysis was carried out on different pig groups using QIIME2 (version 2022.3) [25] to calculate a variety of indexes, including Shannon, Simpson, abundance-based coverage estimators (ACE), and Chao1. To estimate the dissimilarity in the community structure, Bray–Curtis distances were calculated using QIIME2 and visualized using principal coordinates analysis (PCoA).

### 2.8. Microbial Community Structure and Differences

Linear discriminant analysis (LDA) effect size (LEfSe) [27] was used to examine bacterial taxa differentially represented in groups. ASVs with a logarithmic LDA score threshold of >4.0 were selected as showing significant differences between groups.

### 2.9. Function Prediction

The inferred functions of gut microbes have been determined by PICRUSt2 [28]. The predicted genes and their functions were mapped to the Kyoto Encyclopedia of Genes and Genomes (KEGG) database, and group differences were compared using STAMP [29].

### 2.10. Statistical Analysis

Data are expressed as means ± standard deviation (SD). Statistical analysis was performed using GraphPad Prism version 9.0 (San Diego, CA, USA). One-way ANOVA analysis and Wilcoxon rank sum test were employed to assess statistical significance. A significance level of *p* < 0.05 was considered to indicate statistical significance.

## 3. Results

### 3.1. Intestinal Inflammation after LPS Challenge in Piglets

In this study, a total of 20 healthy, weaned Heigai and DLY piglets were intraperitoneally injected with LPS to induce inflammatory responses. As shown in Figure 1A, LPS challenge caused obvious hyperemia in the jejunum of LPS-treated DLY pigs (DLY.LPS); however, this pattern of changes did not appear in that of LPS-treated Heigai pigs (HG.LPS). Microscopically, the integrity of jejunal villi was observed to be destroyed by LPS in both breeds. As shown in Figure 2, LPS challenge significantly reduced the villus height and villus height/crypt depth ratio in both breeds (*p* < 0.001), indicating severe histological damage to the jejunum. The crypt depth increased in both breeds, but only reached statistical significance in DLY pigs, indicating that LPS had a smaller influence on Heigai pigs than on DLY pigs. Additionally, the jejunal villus density was higher in the control group of Heigai pigs (HG.CON) than in the corresponding DLY control group (DLY.CON) (Figure 1B), and the jejunal villus height of HG.CON was lower than that of DLY.CON (*p* = 0.1751) (Figure 2A), suggesting that morphologic differences in the jejunum exist between the two breeds.

Serum concentrations of inflammatory factors, immunoglobulins, and indexes of intestinal permeability and antioxidant capacity, before and after LPS challenge, are shown in Figure 3. LPS challenge elevated the concentrations of TNF-α, IL-10, IFN-γ, IgA, and DAO, and lowered the concentrations of IgM and SOD, in both breeds. Furthermore, the elevations of TNF-α in DLY pigs (*p* < 0.05) and IFN-γ in HG pigs (*p* < 0.01), and the reduction in SOD in HG pigs (*p* < 0.001), reached statistical significance. Additionally, comparison of these factors in the two control groups revealed that HG.CON had significantly higher concentrations of SOD than DLY.CON (26.42 vs. 8.90, *p* < 0.001), demonstrating the higher antioxidant potential of HG pigs.

### 3.2. Analyses of the Jejunal Microbiota by 16S rRNA Gene Sequencing

In total, we obtained 1,926,725 raw paired-end reads, with an average of 96,336 reads per sample. After quality filtering, a total of 1,764,105 clean reads were available, with an average of 88,205 clean reads per sample. The average Phred quality score values of Q20 and Q30 (97.39% and 91.84%, respectively; Appendix A) demonstrated the high quality of sequencing in the study.

Through 16S rRNA (V3-V4 region) amplicon sequencing, a total of 3291 ASVs belonging to the Bacteria domain were identified (Appendix A). As shown in Figure 4A, the rarefaction curves of ASVs tended to be flat at a sequencing depth of 30,000, indicating that the sequencing depth in the study was adequate. The number of ASVs in each sample varied greatly, ranging from 105 to 696 (Figure 4B). Collectively, 81 ASVs were found to be shared across all four groups (Figure 4C). Moreover, the ASV number in HG.CON was much higher than that in DLY.CON, indicating higher species richness within the Heigai pigs.

Approximately 98.63% (3246/3291) of the ASVs identified were annotated in the Silva database (Appendix A). The annotation rates at the phylum, class, order, family, genus, and species levels were 98.63%, 96.63%, 93.41%, 90.98%, 77.04%, and 13.52%, respectively. A total of 35 phyla were identified in the study, among which the most predominant was Firmicutes (relative abundance: 43.54% to 73.56%), followed by Proteobacteria, Cyanobacteria, and Verrucomicrobiota. At the genus level, 469 genera were identified, of which *Lactobacillus*, *Chloroplast*, and *Streptococcus* were the three most abundant genera in both breeds.

The structural composition of microbes was also analyzed. PCoA, performed on the Bray–Curtis dissimilarities, showed significant clustering of the four groups (*p* < 0.001; Figure 5A). There was a large segregation between the bacterial community profiles of the LPS-treated and control piglets. These findings indicated that LPS treatment induces alterations in the structure of the jejunal microbiota in both pig breeds. A large bacterial community difference was also observed between the HG.CON and DLY.CON groups.

### 3.3. Microbial Changes in Response to LPS Challenge

The Shannon and Simpson indexes, which are important indicators of species richness and evenness, decreased after LPS challenge in both breeds (Figure 5B,C), and the Shannon index reached statistical significance (*p* < 0.05) in Heigai pigs. In contrast, the ACE and Chao1 indexes, which mainly reflect microbial species richness, decreased significantly in Heigai pigs (*p* < 0.01) but slightly increased in DLY pigs (Figure 5D,E) after LPS challenge.

We further examined the differences in the 10 most relatively abundant phyla and genera with a relative abundance > 1% among the four groups (Figure 6). At the phylum level (Figure 6C), the relative abundance rates following LPS challenge showed decreases in Firmicutes, Cyanobacteria, Actinobacteriota, Patescibacteria, Fusobacteriota, and Euryarchaeota, but increases in Proteobacteria, Verrucomicrobiota, and Campilobacterota, in both breeds. Additionally, the relative abundance of Bacteroidota was slightly increased in DLY pigs (0.05% vs. 0.21%, *p* = 0.40), but was decreased in Heigai pigs (4.57% vs. 1.06%, *p* = 0.17), following LPS treatment. Actinobacteriota was the only phylum to show a significant (*p* < 0.05) LPS-induced increase in abundance in both breeds. At the genus level (Figure 6D) in both breeds, the relative abundances of *Lactobacillus*, *Escherichia*-*Shigella*, and *Chlamydia* showed increases, while those of *Gemella* and *Saccharimonadales* showed decreases, whereas five other genera exhibited differences between the two breeds in the change trend induced by LPS. Moreover, LPS-induced increases in the abundances of *Escherichia-Shigella* in DLY pigs, and *Gemella* and *Saccharimonadales* in Heigai pigs, reached statistical significance (*p* < 0.05).

Subsequent LEfSe analysis showed significant enrichment in the abundances of seven taxa in DLY.LPS and three taxa in DLY.CON (Figure 7A). At the genus level, DLY.LPS showed enrichment of opportunistic pathogens, including *Escherichia_Shigella* and *Helicobacter*. Notably, 21 taxa were significantly enriched in HG.CON, whereas none were significantly enriched in HG.LPS (Figure 7B). At the genus level, *Gemella*, *Saccharimanadales*, *Clostridia_UCG_014*, and *Dielma* were enriched in HG.CON.

### 3.4. Microbial Difference between Heigai and DLY Pigs

We also compared the microbiota in the two control groups, HG.CON and DLY.CON, which were used to represent the two breeds under normal conditions. HG.CON had significantly higher Shannon, Simpson, ACE, and Chao1 indexes than those of DLY.CON (*p* < 0.05; Figure 5B–E), suggesting that Heigai piglets have higher microbial diversity than DLY piglets.

At the phylum level, the relative abundances of Verrucomicrobiota, Patescibacteria, Bacteroidota, Fusobacteria, and Euryarchaeota were higher in HG.CON than in DLY.CON, whereas the relative abundances of Proteobacteria, Cyanobacteria, and Actinobacteriota were lower in HG.CON than in DLY.CON (Figure 6C). Furthermore, between-group abundance differences in Verrucomicrobiota, Patescibacteria, and Bacteroidota were significant (*p* < 0.05). At the genus level, the relative abundances of *Gemella*, *Saccharimonadales*, *Clostridia_UCG_014*, *Terrisporobacter*, *Dielma Turicibacter*, *Christensenellaceae_R-7_group RF39*, and *Prevotella* were significantly higher in HG.CON than in DLY.CON (*p* < 0.05), whereas that of *Sarcina* was significantly lower in HG.CON than in DLY.CON (*p* < 0.05; Figure 6D).

Further LEfSe analysis of the two control groups demonstrated that 27 taxa were significantly enriched in HG.CON and 13 taxa were enriched in DLY.CON (Figure 7C). At the genus level, we observed that *Gemella*, *Saccharimanadales*, *Dielma*, *Clostridia_UCG_014*, and *Terrisporobacter* were significantly more preponderant in HG.CON, whereas *Mitochondria*, *Chloroplast*, and *Sarcina* were more abundant in DLY.CON.

### 3.5. Differences in Functional Potential of Jejunal Microbial Communities

To better elucidate functional changes induced by the LPS challenge, functions of the jejunal microbiota were predicted at KEGG taxonomy level 3 by PICRUSt2. Comparisons of the LPS and control groups in Heigai pigs revealed significant differences in 23 pathways (Figure 8A). Microbial gene functions involved in pathways, including amino acid metabolism (e.g., histidine metabolism, and valine, leucine, and isoleucine degradation), carbohydrate metabolism (pentose and glucuronate interconversions), energy metabolism (carbon fixation pathways in prokaryotes, oxidative phosphorylation), and degradation of harmful compounds (styrene degradation), were significantly more prevalent in HG.CON (*p* < 0.05), whereas pathways regulating the phosphotransferase system (PTS) and metabolic disease (type I and type II diabetes mellitus, renal cell carcinoma) were significantly more prevalent in HG.LPS (*p* < 0.05). Between the two DLY groups, only one pathway (proteasome) was significantly more prevalent in DLY.CON than in DLY.LPS (*p* < 0.05; Figure 8B).

A batch of pathways with significant differences between the two control groups was also identified (Figure 8C). The microbial gene functions found to be higher in Heigai pigs were involved in pathways including the following: essential amino acid metabolism (e.g., lysine degradation and biosynthesis, and valine, leucine and isoleucine degradation); carbohydrate metabolism (citrate cycle, pentose and glucuronate interconversions, propanoate metabolism); and biosynthesis of important compounds (pantothenate and CoA), vitamins (riboflavin), antibiotics (novobiocin, ansamycins), and alkaloids (isoquinoline alkaloid, tropane, piperidine, pyridine). Pathways found to be higher in DLY pigs were related to amino acid metabolism (tyrosine, D-glutamine and D-glutamate, glutathione), biosynthesis of important compounds (inositol phosphate metabolism, retinol metabolism, polyketide sugar unit), and xenobiotic degradation and metabolism (e.g., dioxin, xenobiotics by cytochrome P450, naphthalene).

## 4. Discussion

Intestinal inflammation is the most common acute intestinal infectious disease in piglets, inducing intestinal mucosal injury, microbiota dysbiosis, and diarrhea. Among the six segments of the gut, the jejunum is the major site of nutrient digestion and absorption, and is severely affected by diarrhea [30]. Here, we provided a comprehensive comparative investigation of Heigai and DLY piglets, analyzing the effects of LPS challenge on the microbial profiles of jejunal communities and the jejunal mucosa, screening for bacteria associated with diarrhea, and exploring the special microbial characteristics of the Heigai pig breed.

### 4.1. Effect of LPS Challenge on Serum Indexes and Jejunal Mucosa

Villus height, crypt depth, and the villus height/crypt depth ratio are important indicators for assessing intestinal function and health. In this study, we found that LPS-treated pigs had lower villus height and villus height/crypt depth ratio, and higher crypt depth, compared with control pigs (Figure 2), indicating that LPS caused severe intestinal damage leading to diarrhea. These results are in accordance with most of the published studies on the effects of LPS on intestinal morphology [9,31,32]. Furthermore, the key cytokines and indexes of intestinal permeability and antioxidant capacity also exhibited changes after LPS challenge. Our findings showed that LPS challenge led to elevated serum levels of IFN-γ, TNF-a, IL-10, and DAO, and lower levels of SOD, compared with untreated controls (Figure 3), which are in good agreement with the results of previous studies on mice [33], pigs [9], lambs [34], and chickens [35]. Taken together, the evidence indicates that LPS is a suitable agent to mimic the bacterial inflammatory response in weaned pigs.

### 4.2. Effects of LPS Challenge on the Composition and Functions of the Jejunal Microbiota

Previous studies revealed that a wide range of gastrointestinal diseases and metabolic disorders are accompanied by variations in the composition and functions of the gut microbiota [13,36]. Our findings corroborated that LPS impacts the composition and richness of the microbial population, causing partial alterations in the structure of the gut microbiota in both pig breeds (Figure 5). Firmicutes and Proteobacteria, accounting for 74% of the total bacterial abundance, were the two major phyla in the jejunum of both breeds (Figure 6A). After LPS challenge, the relative abundance of Firmicutes decreased, and that of Proteobacteria increased. Previous studies [9,37] measured microbial profile changes in the ileum instead of the jejunum. However, while changes in the abundance of Firmicutes were observed to be similar to our study, the abundance of Proteobacteria accounted for a very small proportion of the total abundance in the ileum in previous studies, with almost no changes observed, possibly due to the different intestinal segments used. At the genus level, the relative abundance of *Escherichia-Shigella* increased considerably from 0.24% to 12.01% in DLY pigs and from 0.50% to 9.99% in Heigai pigs. This conditional intestinal pathogen has a symbiotic relationship with the host, but its significant increase under LPS challenge may be harmful, inducing inflammation and diarrhea. Additionally, the relative abundances of *Gemella* and *Saccharimonadales* significantly decreased in Heigai pigs (*p* < 0.05), but only slightly decreased in DLY pigs, after LPS challenge. Although there are no reports of ill effects on animals, these bacteria have been associated with human inflammatory mucosal diseases [38], and allergies and asthma in children [39]. Therefore, these three genera may induce distinctly different inflammatory responses in the two breeds.

We also discovered that LPS challenge induced important functional changes in the jejunal bacterial community in Heigai pigs. Pentose and glucuronate interconversions, carbon fixation pathways in prokaryotes, oxidative phosphorylation, and styrene degradation were all significantly decreased. Three of these categories (pentose and glucuronate interconversions, carbon fixation pathways, and oxidative phosphorylation) are carbohydrate and energy metabolism-related pathways. Styrene, one of the most important monomers produced by the chemical industry today, is used as a sole source of carbon and energy by a large number of microorganisms [40]. Thus, LPS challenge triggers a decrease in energy metabolism in the jejunal bacterial community. By contrast, LPS challenge also induced significant increases in functions related to the PTS, type I and type II diabetes mellitus, and renal cell carcinoma. PTS is the main system involved in sugar transportation and phosphorylation, but has recently been shown to be associated with bacterial resistance to stress factors [41]. Type I and type II diabetes mellitus and renal cell carcinoma are well-known disease-related pathways. Thus, LPS challenge also triggers an increase in bacterial activity related to the stress response. These findings indicate that LPS challenge triggers functional changes in energy metabolism and bacterial activity related to the stress response in the jejunal community of Heigai pigs, and may partly explain the comparatively weaker inflammatory response in these pigs.

### 4.3. Characteristics of Immune Response and Intestinal Microbiota of Heigai Pigs

Heigai is a typical indigenous pig breed of China, characterized by roughage tolerance and a low rate of diarrhea in weaning piglets. In Heigai pigs undergoing LPS challenge, the jejunal hyperemia was less obvious than that in DLY pigs (Figure 1A), with smaller changes in jejunal villus height and crypt depth, indicating that Heigai pigs experienced less intestinal mucosal damage than DLY pigs. Additionally, although LPS induced a decrease in the SOD level in both breeds, HG.LPS still had a significantly higher SOD level than DLY.LPS (*p* < 0.001). SOD is an important antioxidant enzyme that plays a key role in diminishing oxidative stress, an essential pathological process in LPS-induced inflammatory responses. Taken together, these findings indicate that Heigai piglets have a weaker immune response to LPS challenge than DLY piglets, which may be one reason for their lower rate of diarrhea in the weaning stage.

Previous studies have shown that there are differences in the composition and functions of the gut microbiota among pig breeds [18,42]. In our study, we found that the relative abundances of *Gemella*, *Saccharimonadales*, *Clostridia_UCG_014*, *Terrisporobacter*, and *Dielma* were significantly higher in HG.CON than in DLY.CON, and were significantly enriched in HG.CON in LEfSe analysis. Most *Clostridia* species have a commensal relationship with their host, playing a crucial role in gut homeostasis by releasing butyrate as an end-product of fermentation [43]. Butyrate inhibits the activation of the transcription factor nuclear factor-κB, leading to an anti-inflammatory effect [44]. *Terrisporobacter* produces acetic acid from simple carbohydrates [45] and has been shown to play important roles in the organic material degradation of compost [46]. *Dielma* is reportedly positively correlated with the production of short-chain fatty acids in the gut microbiota of broilers [47]. Similarly, Heigai pigs possess a unique bacterial composition that may be related to their special phenotypic features.

Apart from the differences in bacterial composition, we found many functional pathways with significantly different proportions between the two control groups. Specifically, pathways related to essential amino acids, such as lysine degradation and biosynthesis, and valine, leucine and isoleucine degradation, were more prevalent in Heigai pigs, whereas those related to nonessential amino acids, such as tyrosine, D-glutamine and D-glutamate, and glutathione, were more prevalent in DLY pigs. The proportions of pathways related to carbohydrate metabolism, including the citrate cycle, pentose and glucuronate interconversions, and propanoate metabolism were significantly higher in Heigai pigs. The biosynthesis of antibiotics (novobiocin, ansamycins) and alkaloids (isoquinoline, tropane, piperidine, pyridine) was also higher in Heigai pigs. According to previous studies, these metabolites have antibacterial, anticancer, and immune regulatory activities [48,49,50]. Therefore, intestinal microbiota play crucial roles in the amino acid metabolism, energy maintenance, and stress response of Heigai pigs.

## 5. Conclusions

LPS successfully induced bacterial inflammatory response in the experimental pigs, changed the microbial composition, and triggered functional changes in energy metabolism and activities related to the stress response in the jejunal bacterial community. This study also revealed that Heigai pigs had a weaker immune response to LPS challenge than DLY pigs, and identified several genera related to the breed-specific phenotypes of Heigai pigs. This study comprehensively compared the effects of LPS challenge on jejunal mucosal morphology, jejunal microbial composition, and serum indexes in two pig breeds, and would be beneficial to discover the mechanism of the inflammatory response in piglet diarrhea.

## Figures and Tables

**Figure 1 animals-14-00665-f001:**
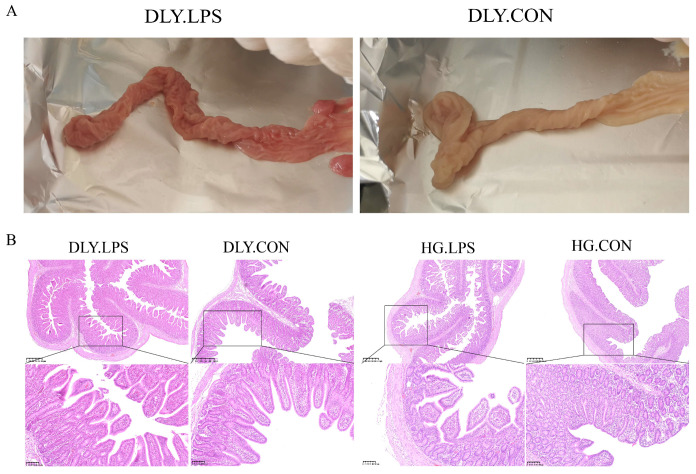
Jejunal inflammation after LPS challenge. (**A**) Comparison of jejunal segments between LPS-treated DLY (DLY.LPS) and control DLY (DLY.CON) group pigs. (**B**) Representative H&E staining images of jejunal sections of DLY.LPS, LPS-treated Heigai (HG.LPS), DLY.CON, and control Heigai (HG.CON) pigs under the digital pathology section scanner.

**Figure 2 animals-14-00665-f002:**
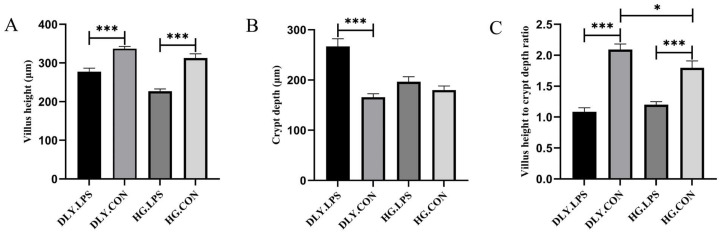
Effects of LPS on jejunal mucosal villus height (**A**), crypt depth (**B**), and the villus height/crypt depth ratio (**C**). Significance is presented as * *p* < 0.05, *** *p* < 0.001, as determined by one-way ANOVA analysis. Data expressed as means ± SD. DLY.LPS represents LPS-treated DLY pigs, DLY.CON represents control DLY pigs, HG.LPS represents LPS-treated Heigai pigs, and HG.CON represents control Heigai pigs.

**Figure 3 animals-14-00665-f003:**
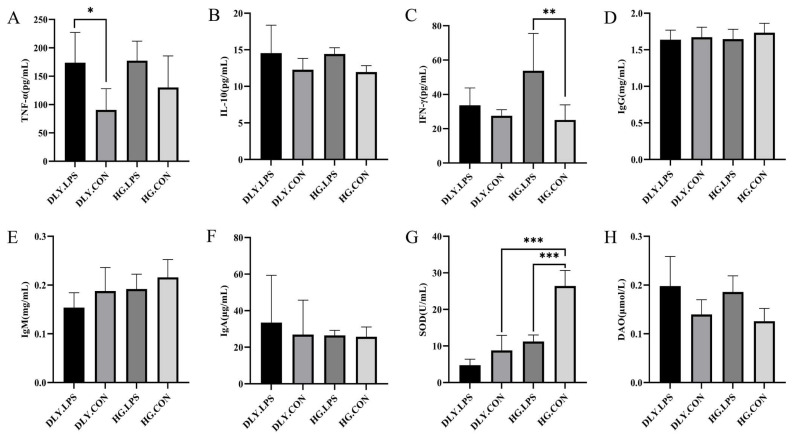
Effects of LPS on serum indexes of (**A**) TNF-α, (**B**) IL-10, (**C**) IFN-γ, (**D**) IgG, (**E**) IgM, (**F**) IgA, (**G**) SOD, and (**H**) DAO. Significance is presented as * *p* < 0.05, ** *p* < 0.01, *** *p* < 0.001, as determined by one-way ANOVA analysis. Data expressed as means ± SD. DLY.LPS represents LPS-treated DLY pigs, DLY.CON represents control DLY pigs, HG.LPS represents LPS-treated Heigai pigs, and HG.CON represents control Heigai pigs.

**Figure 4 animals-14-00665-f004:**
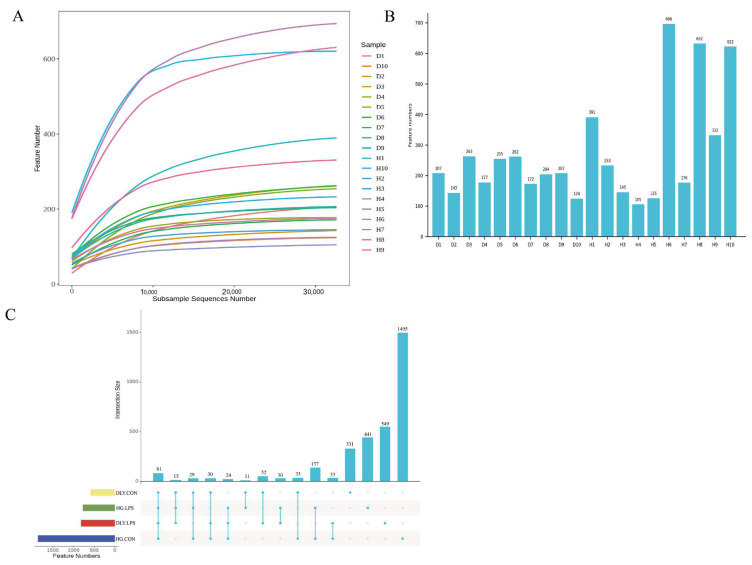
Rarefaction curves of ASVs and ASV distribution in individual pigs and in the four groups. (**A**) Rarefaction curves of ASVs in the jejunal samples. (**B**) Histogram of the number of ASVs in each sample. (**C**) UpSet plot of the number of ASVs among groups. D1 to D5 are sample ID of LPS-treated DLY pigs. D6 to D10 are sample ID of control DLY pigs. H1 to H5 are sample ID of LPS-treated Heigai pigs. H6 to H10 are sample ID of control Heigai pigs. DLY.LPS represents LPS-treated DLY pigs, DLY.CON represents control DLY pigs, HG.LPS represents LPS-treated Heigai pigs and HG.CON represents control Heigai pigs.

**Figure 5 animals-14-00665-f005:**
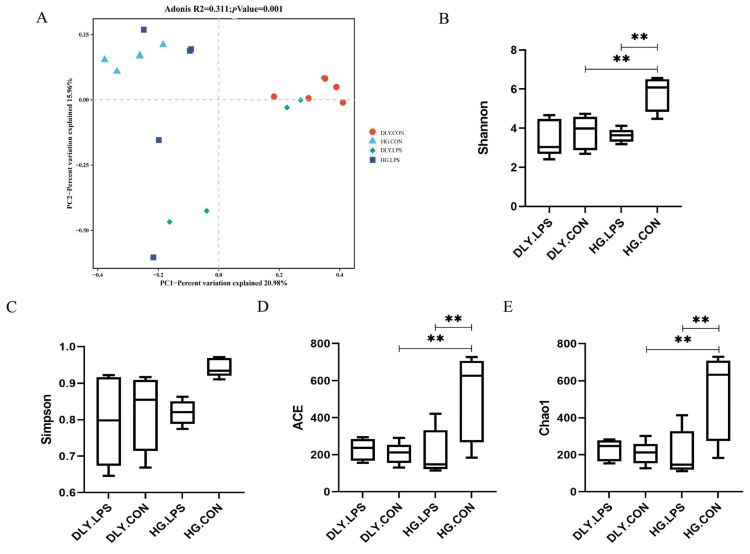
Effects of LPS on the alpha and beta diversity of jejunal microbiota. (**A**) PCoA depicting the distribution of samples according to the Bray–Curtis distance. (**B**–**E**) Comparison of alpha diversity using the (**B**) Shannon index, (**C**) Simpson index, (**D**) ACE index, and (**E**) Chao1 index. All data expressed as means  ±  SD. Significant mean difference are indicated with ** for *p* < 0.01, as determined by one-way ANOVA analysis. DLY.LPS represents LPS-treated DLY pigs, DLY.CON represents control DLY pigs, HG.LPS represents LPS-treated Heigai pigs, and HG.CON represents control Heigai pigs.

**Figure 6 animals-14-00665-f006:**
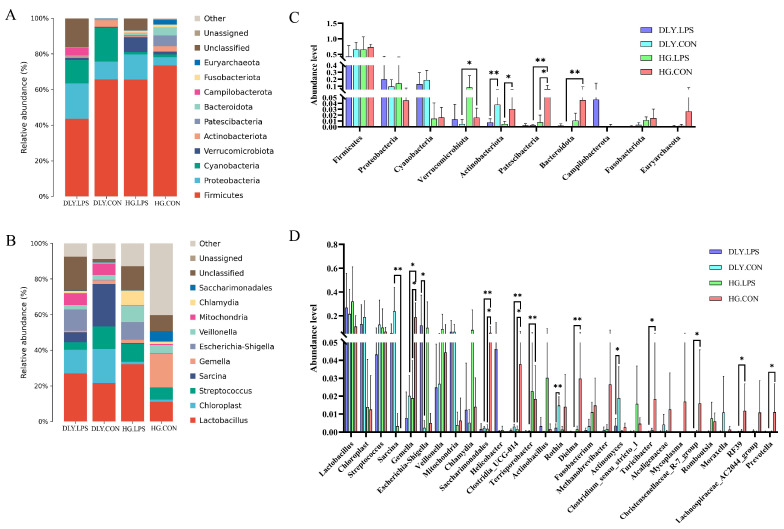
Effects of LPS on the relative abundances of intestinal microbes in Heigai and DLY pigs. (**A**) Histogram of the 10 most relatively abundant phyla. (**B**) Histogram of the 10 most relatively abundant genera. (**C**) Comparison of the 10 most relatively abundant phyla among the groups. (**D**) Comparison of genera with relative abundance >1% among the groups. Significant mean differences evaluated by Wilcoxon rank sum test are indicated with * and ** for *p* < 0.05 and *p* < 0.01, respectively. DLY.LPS represents LPS-treated DLY pigs, DLY.CON represents control DLY pigs, HG.LPS represents LPS-treated Heigai pigs, and HG.CON represents control Heigai pigs.

**Figure 7 animals-14-00665-f007:**
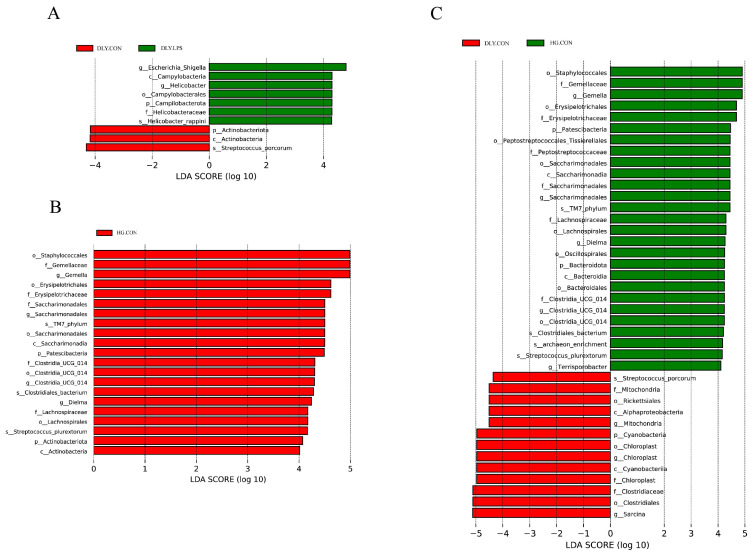
Bacterial taxa differentially presented in each group identified by LEfSe based on LDA score threshold >4.0. (**A**–**C**) Bacterial taxa with a differential presence between (**A**) DLY.CON and DLY.LPS, (**B**) HG.CON and HG.LPS, and (**C**) DLY.CON and HG.CON. DLY.LPS represents LPS-treated DLY pigs, DLY.CON represents control DLY pigs, HG.LPS represents LPS-treated Heigai pigs, and HG.CON represents control Heigai pigs.

**Figure 8 animals-14-00665-f008:**
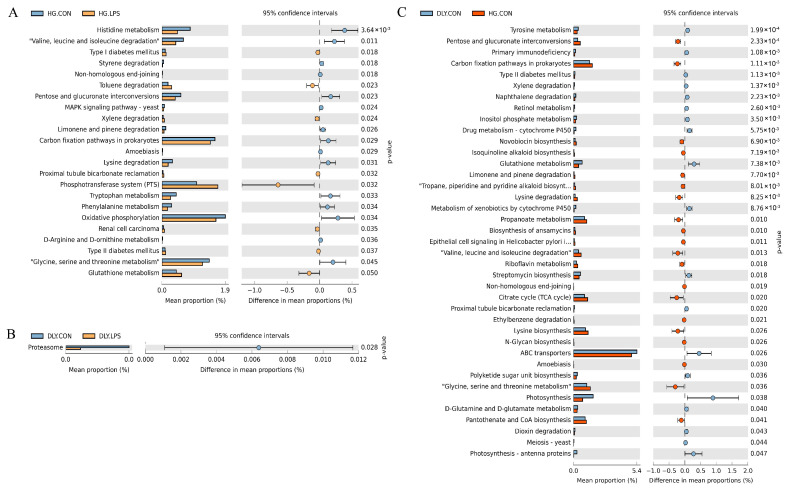
Differences in the predicted functions of the jejunal microbiota among the groups. Differences in jejunal microbial functions between (**A**) HG.CON and HG.LPS, (**B**) DLY.CON and DLY.LPS, and (**C**) DLY.CON and HG.CON. DLY.LPS represents LPS-treated DLY pigs, DLY.CON represents control DLY pigs, HG.LPS represents LPS-treated Heigai pigs and HG.CON represents control Heigai pigs.

**Table 1 animals-14-00665-t001:** Ingredient composition and nutritional levels of the diets.

Ingredients	Diets, %
Puffed corn	56.5
Fish meal	5
Whey powder	2.5
Puffed soybean meal	23.5
Rice bran meal	1
Glucose	5
Soybean	10.0
Solidified fat	1.5
Acidifying agent	0.3
Lysine	0.5
Methionine	0.2
Concentrate	4
NaCl	0.4
Total	100
Nutrient levels	
Digestible energy (MJ/kg)	14.00
Crude protein (%)	20.00
Lysine (%)	1.45
Total calcium (%)	0.60
Total phosphorus (%)	0.50
Crude fiber (%)	1.77

Provided per kilogram of concentrate: 20 mg Mn, 200 mg Fe, 160 mg Zn, 20 mg Cu, 1.5 mg I, and 1 mg Se; 10,000 IU vitamin A; 4000 IU vitamin D; 80 mg vitamin E; 20 mg vitamin B_2_; 2.5 mg vitamin K_3_ and 0.05 mg vitamin B_12_. Crude protein, crude fiber, total calcium, and total phosphorus were analyzed values, the rest were calculated values.

## Data Availability

Sequence data have been deposited in the NCBI database and assigned accession number PRJNA1025323.

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
