# Peer review of "Comparative Analysis of Intestinal Inflammation and Microbiota Dysbiosis of LPS-Challenged Piglets between Different Breeds"

_animals, 2024, doi:10.3390/ani14050665_

Round 1

Reviewer 1 Report

Comments and Suggestions for Authors

This manuscript was about the breed difference of pigs in responses of intestinal inflammation and microbiota to LPS challenge.  This is a nice study, and the results are meaningful. The experimental design and methodology were described clearly, and the manuscript was organized and presented well.

Minor

Materials and method, the average body weight and genders for each group need to be reported.

Statistical analysis, I would suggest analyzing the data following the two-way ANOVA with the F-test.  This is a typical 2 (breeds) x 2 (±LPS) factorial design. If the interaction is not significant, you may perform one-way ANOVA analysis but not a t-test.  “t” test is not proper test for this study.

In Figure 3 A and C, the authors may need to check the statistical analysis. The error bars for the significance look overlaid, indicating the two columns may not be significant.

Reviewer 2 Report

Comments and Suggestions for Authors

In this study, the author In the current study, we performed a comprehensive comparative analysis of the effects of LPS challenge on jejunal mucosal morphology, jejunal microbial composition, and serum indexes in two pig breeds: DLY and Heigai, an indigenous Chinese breed. This finding uncovered differences between Heigai and DLY pigs in intestinal inflammation and microbiota dysbiosis induced by the LPS challenge, providing a theoretical basis for unraveling the mechanism of intestinal inflammation in swine and proposing microbial candidates involved in the resistance to diarrhea in piglets. However, the readability of the manuscript can also be greatly improved. Through editing and some modifications, I think this manuscript will be more suitable for publication.

1. The number of Abstract words should be limited to 250 words. The author should condense the content.

2. The Introduction section, the description of LPS should include the research progress at home and abroad. Please add relevant content and specific references.

3. Introduction “This study is the first to apply an LPS-induced inflammation model to investigate and compare the breed-specific factors leading to intestinal inflammation in piglets, and may contribute to future screening for intestinal microbes associated with this common illness.” Please use the word "first" with caution. Hasn't it been reported before? If this is the first animal model, please add more evidence.

4. The Material and Method section should include detailed “material”. The author should complete it.

5. There are many inconsistent picture descriptions in the article, such as “Figure 1A”, “Fig. 2,”, “Fig.3.” and so on. Authors should unify the full-text format.

6. In Fig. 4, Fig. 5A, Fig. 6, Fig. 7 and Fig. 8, the pixels in the pictures are not clear. Please provide clear result Figures.

7. The reference list should primarily include research publications from the past three years. However, this manuscript has a total of 50 references and contains more than 50% of references from before 2020. The author should replace the references and standardize the format.

Comments on the Quality of English Language

Moderate editing of English language required

Reviewer 3 Report

Comments and Suggestions for Authors

In production, we found that the rate of diarrhea was lower in weaned Heigai piglets than in Duroc × Landrace × Yorkshire (DLY) crossbred piglets, which is the most widely used commercial pig crossbred worldwide. – please provide the references

Statistical analysis –please specify which data were analyzed with which method

Figure 4,  5a, 6, 7 - please provide better resolution

“In contrast, the ACE and Chao1 indexes, which mainly reflect microbial species richness, increased in DLY pigs but decreased in Heigai pigs (P < 0.05) (Fig. 5D, E) after LPS challenge” the sentence is misleading  - for DLY pigs the changes are not significant according to figure 5 D, E

“These results indicated that the two breeds had different responses to LPS challenge in terms of microbial species richness, but similar responses to LPS challenge when both species richness and evenness were consid-ered.” Since the results for DLY are not significant we can not say that the responses were similar.

 The most common bacteria in pig gut are Firmicutes and Bacteroidota, while Bacteroidota were very rare in your data – what is the possible reason? Ratio of Firmicutes to Bacteroidota is considered as an indicator of gut health – it could be included in the analysis.
